# Diversity of Endophytic Fungi Isolated from *Prunus yedoensis* and Their Antifungal Activity Against Wood Decay Fungi

**DOI:** 10.3390/microorganisms13030617

**Published:** 2025-03-07

**Authors:** Misong Kim, Manh Ha Nguyen, Sanggon Lee, Wonjong Han, Minyoung Kim, Hyeongguk Jeon, Jinheung Lee, Sangtea Seo, Namkyu Kim, Keumchul Shin

**Affiliations:** 1Department of Forest Environmental Resources, College of Agriculture and Life Sciences, Gyeongsang National University, Jinju 52828, Republic of Korea; misoong427@naver.com (M.K.); tkdrhs170@naver.com (S.L.); gksdnjsb@naver.com (W.H.); 2Institute of Agriculture & Life Science, Gyeongsang National University, Jinju 52828, Republic of Korea; manhhafsiv@gmail.com (M.H.N.); mume@gnu.ac.kr (M.K.); naviflower1@gnu.ac.kr (H.J.); 3Forest Protection Research Center, Vietnamese Academy of Forest Sciences, Hanoi 11910, Vietnam; 4Department of Forest Healing, College of Human Service, Catholic Kwandong University, Gangneung 25601, Republic of Korea; ghost11010@naver.com; 5Division of Forest Diseases and Insect Pests, National Institute of Forest Science, Seoul 02455, Republic of Korea; stseo@korea.kr

**Keywords:** biocontrol, culture filtrate, endophytes, *Fusarium acuminatum*, white rot fungi

## Abstract

Endophytic fungi play a vital role in protecting and promoting the growth of their host plants. The diversity of fungal endophytes has been documented across different host plant species and varies depending on factors such as the species of the host, ecological conditions, and the health status of the plant. We isolated endophytic fungi from *Prunus yedoensis* trees with different decay rates. A total of 31 species were isolated from decayed trees, while 33 species were obtained from healthy trees. The number of endophytic fungi exhibiting antifungal activities against wood decay fungi was higher in healthy trees, with 10 species showing activity compared to only 1 species from decayed trees. Endophytic fungus *Fusarium acuminatum* (BEN48) had the highest inhibition rates against *Trametes versicolor*, *Ganoderma gibbosum*, and *Vanderbylia fraxinea*. Heating conditions did not significantly affect the inhibitory ability of the culture filtrate from BEN48 on wood decay fungi. At 50% concentration, the inhibitory abilities of the culture filtrates against *Trametes versicolor*, *Ganoderma gibbosum*, and *Vanderbylia fraxinea* were 96.5%, 64.1%, and 92.7%, respectively. The inhibitory effects against *Trametes versicolor* decreased at concentrations of 30% and 10%, resulting in inhibition rates of 83.7% and 50.8%, respectively. For *Ganoderma gibbosum*, the inhibition rate reduced to 52.6% at 30% concentration and 24.5% at 10% concentration. For *Vanderbylia fraxinea*, there was no significant difference between the 30% and 10% concentrations, and the inhibition rates for both concentrations were high, measuring 89.9% and 88.8%, respectively. Hence, *Fusarium acuminatum* (BEN48) has promise as a biocontrol agent for managing wood decay fungi.

## 1. Introduction

Fungal endophytes can inhabit various plant tissues, exhibiting high species richness across host tree species [1,2,3]. Each tree species can host one or even several hundred endophytic fungal isolates [4]. Endophytic fungi are diverse in both coniferous and broad-leaved trees [3,5]. The diversity of endophytic fungi is greater in forest trees than in other plants [6]. However, the dominant species of fungal endophytes varies among different host trees and is influenced by geography, tree species, environmental conditions, tree genotypes, and tree age [3,6,7]. Additionally, the number of endophytic fungal isolates obtained can depend on the surface sterilization method used, the size of the tissue samples collected, and the types of culture media employed [8,9]. Endophytic fungi are an effective biocontrol agent and play an important role in controlling diseases in agriculture and forestry [10,11]. For example, *Daldinia childiae* and *Alternaria alternata* isolated from *Quercus* species showed antifungal activities against oak wilt disease caused by *Raffaelea quercus-mongolicae* [12]. This pathogen was also inhibited by the culture filtrate of endophytic fungus *Nectria balsamea* isolated from *Pinus densiflora* [12,13]. Several fungal endophytes can produce compounds that have antifungal properties and suppress plant pathogens [14]. Pestalachlorides A (C_21_H_21_Cl_2_NO_5_) and B (C_20_H_18_Cl_2_O_5_) were compounds produced by the endophytic fungus *Pestalotiopsis adusta*. These two compounds could inhibit popular plant pathogens *Verticillium albo-atrum*, *Gibberella zeae*, and *Fusarium culmorum* [15]. The endophytic fungus *Chaetomium globosum* and *Phomopsis* species isolated from *Ginkgo biloba* produced Cytosporones B and C and Chaetomugilins A and D, which exhibited antifungal activities against *Candida albicans* and *Fusarium oxysporum* [16].

*P. yedoensis* has been widely grown in several Northeast Asian countries, including South Korea [17,18,19,20]. This species is commonly cultivated in urban areas and produces beautiful blooms in the spring [17,21]. Research on tissue cultures of *P. yedoensis* and the effects of technical measures on the rooting and growth of transplanted cuttings has been conducted in South Korea [22,23,24]. Furthermore, the leaf extract from this tree demonstrated various biological activities, including antimicrobial properties, the crystallization of silver metal, and the removal of phosphate [25,26,27,28]. However, *P. yedoensis* trees have been observed to show symptoms of trunk decay caused by various white rot fungi, mainly including *Ganoderma gibbosum*, *Trametes versicolor*, and *Vanderbylia fraxinea* [29,30]. Several decayed trees may die while standing or could fall, posing risks in urban areas. Research on the control of these fungi is still limited in South Korea. The antifungal activity of actinomycete strains against brown rot fungi was evaluated through screening [31]. While the use of endophytic fungi for the biological control of phytopathogens has been explored for some pathogens, there has been no investigation into selecting endophytic fungal isolates from *P. yedoensis* trees that exhibit antifungal activity against these white rot fungi. Therefore, this study was conducted to isolate and select endophytic fungal isolates with antifungal activity in both dual culture and culture filtrate experiments and to evaluate the effects of different concentrations on antifungal activity against wood decay fungi.

## 2. Materials and Methods

### 2.1. Isolation of Endophytic Fungi

Endophytic fungi were isolated from wood tissue samples collected from both healthy and decayed *P. yedoensis* planted as urban street trees in Gyeongnam province in 2022~2023. The healthy and decayed trees were determined after measuring the internal defect rate at the lower part of the stem using a sonic tomograph. Wood tissue samples (100 wood drilling fragments) were collected from 5 healthy trees with a 0~1% defect rate and 5 decayed trees with a 35~45% defect rate using auger drilling. The samples were gently washed under tap water and then surface-sterilized using a 2% sodium hypochlorite (NaOCl) solution for 5 min. This was followed by immersion in 70% ethanol for 3 min, after which they were rinsed three times with sterile distilled water, each rinse lasting 15 s. Once the surface was sterilized, the samples were dried to prepare for the isolation of endophytic fungi. They were cut into small pieces measuring 5 × 5 × 8 mm using sterile tools and placed on Petri dishes containing Potato Dextrose Agar (PDA) with antibiotics (100 mg/L streptomycin). The agar plates were incubated at 25 °C for 7 to 10 days until mycelial growth emerged from the samples. To purify the endophytic fungi, mycelial growth tips or single spores were cultured onto new PDA plates. Subcultures were conducted multiple times until completely pure isolates were obtained. The pure isolates were then coded and stored in agar slants at 4 °C for further studies.

### 2.2. Identification of Endophytic Fungi

Endophytic fungi were classified based on morphological characteristics such as colony appearance, color, mycelial growth, and sporulation structures, using both the naked eye and a microscope (Nikon Eclipse Ci-L, Tokyo, Japan). Genomic DNA was extracted using the Quick-DNA Fungal/Bacterial Miniprep Kit (ZYMO RESEARCH, Tustin, CA, USA). Mycelia from representative endophytic fungi, identified based on their morphological characteristics, were used for DNA extraction. The internal transcribed spacer (ITS) region of the ribosomal DNA (rDNA) was then amplified by a polymerase chain reaction (PCR) using a T-100 thermocycler from Biorad, Hercules, CA, USA. To amplify the ITS region, a combination of primers ITS1F (5′-CTTGGTCATTTAGAGGAAGTAA-3′) and ITS4 (5′-TCCTCCGCTTATTGATATGC-3′) was utilized [32]. The PCR conditions involved an initial denaturation step at 95 °C for 3 min, followed by 40 cycles of denaturation at 95 °C for 30 s, annealing at 54 °C for 50 s, and extension at 72 °C for 1 min. A final extension was performed at 72 °C for 5 min, after which the samples were stabilized and stored at 4 °C. The presence or absence of the amplified PCR product was confirmed using electrophoresis on a 1.5% agarose gel. The PCR product was subsequently purified with a DNA purification kit (Exonuclease, Thermo Scientific, Waltham, MA, USA). Macrogen Co., Ltd. performed the base sequence analysis using an ABI 3730xl autosequencer (Applied Biosystems, Waltham, MA, USA). The resulting sequence was edited with Chromas version 2.6.2 and BioEdit version 7.2.5, and the edited sequence was then identified as a similar species through a Nucleotide BLAST search in the National Center for Biotechnology Information (NCBI) database (https://blast.ncbi.nlm.nih.gov/Blast.cgi, 21 January 2025).

### 2.3. Testing for Antifungal Activities of Isolated Endophytic Fungi

For the primary screening, 257 isolates were used to test for antifungal activities against three wood decay fungi which were isolated from *P. yedoensis*, namely *Ganoderma gibbosum* (OR570866), *Trametes versicolor* (MN294847), and *Vanderbylia fraxinea* (OQ612670.1). These wood decay fungi were subcultured from agar slants that had been stored at 4 °C. A cork borer was used to take out mycelial plugs from endophytic fungi and the test pathogens. Four different endophytic fungal isolates were cultured on a PDA plate, equidistant and near the periphery. A mycelial plug of each pathogen was placed on the center of the PDA plate. All plates were incubated in the dark at 25 °C for 7 days to inspect the antifungal activities of endophytic fungal isolates. Inhibition zones were measured as distances between the colonies of the test fungus and endophytic fungi. Eleven endophytic fungal isolates with an inhibition zone above 1.0 mm were selected for the secondary screening. Endophytic fungi and test pathogens were cultured on opposite sides of PDA Petri dishes; only test pathogens were cultured without endophytic fungi for the control treatment. The mycelial growth inhibition (MGI) of endophytic fungi against test pathogens was calculated using the following formula:$$MGI\left(\%\right)=\frac{C-T}{C}\times 100\%$$
where C is the mycelial growth of the test pathogen in the control plate, and T is the mycelial growth of the test pathogen in the treated plate.

### 2.4. Antifungal Activities of Culture Filtrates from Selected Endophytic Fungi

Three endophytic fungal isolates with the highest inhibition rates against each pathogen were selected for the culture filtrate test. A total of five isolates were chosen for this test: BEN07, BEN36, BEN37, BEN48, and BEN97. Culture filtrates were prepared, and experiments were conducted following the method of a previous study [12]. The culture filtrates were mixed with PDA medium (1:1) and tested under both heated (autoclaved at 121 °C for 15 min) and unheated (not autoclaved) conditions. The inhibition rates were measured based on the mycelial growth of the test pathogens in both the treatments and controls, and these rates were calculated using the formula mentioned above.

BEN48 was selected to assess the effects of culture filtrate concentration on the mycelial growth of wood decay fungi. The culture filtrate of BEN48 was mixed with Potato Dextrose Agar (PDA) at three concentrations: 10%, 30%, and 50%. The assessment of antifungal activity at different concentrations was conducted in a manner similar to the previous experiment.

### 2.5. Statistical Analysis

A one-way analysis of variance (ANOVA) was conducted, followed by Tukey’s honestly significant difference (HSD) test as the post hoc analysis, to examine the differences in antifungal activities against wood decay fungi among endophytic fungal isolates at a 5% significance level. All statistical analyses were performed using IBM SPSS Statistics version 27 for Windows.

## 3. Results

### 3.1. Diversity of Endophytic Fungi Isolated from P. yedoensis

The diversity of endophytic fungi isolated from *P. yedoensis* is summarized in Table 1. A total of 257 endophytic fungal isolates were collected, with 184 sourced from decayed trees and 73 from healthy trees. In total, 49 species were identified from all isolates, comprising 16 species that were found exclusively in decayed trees, 18 species that were found exclusively in healthy trees, and 15 species that were present in both decayed and healthy trees (Figure 1). The frequency of *Parapyrenochaeta maryellenpeartiae* was the highest in decayed trees, occurring at a rate of 35.87%, compared to 21.92% in healthy trees. The second most frequent species in decayed trees was *Paraboeremia putaminum*, at 11.96%, followed by *Paraconiothyrium brasiliense*, which accounted for 9.24%. In healthy trees, the frequencies were similar for *Paraconiothyrium brasiliense* at 9.59% and *Pyrenochaeta* sp. at 8.22% (Table 1). Several fungal species were identified only once during the isolation process, with a frequency of occurrence of 0.54% in decayed trees and 1.37% in healthy trees such as *Allophoma zantedeschiae*, *Aspergillus sydowii*, *Cladosporium perangustum*, *Cladosporium ramotenellum*, and *Nigrograna acericola* (Table 1).

### 3.2. Screening of Endophytic Fungi Against Wood Decay Fungi in Dual Culture Test

The evaluation results of endophytic fungal isolates with antifungal activity against wood-decaying fungi are presented in Table 2 and illustrated in Figure 2. Eleven isolates demonstrated antifungal activity against three wood decay fungi: *Trametes versicolor* (FP268), *Ganoderma gibbosum* (FP281), and *Vanderbylia fraxinea* (FP287). Among 11 antifungal isolates, only 1 was obtained from decay trees, while the others were sourced from healthy trees. The mycelial growth inhibition rates of the endophytic fungi against *Trametes versicolor* ranged from 12.59% to 40.27%. The inhibition rates against *Ganoderma gibbosum* ranged from 12.50% to 47.51%, while the rates against *Vanderbylia fraxinea* ranged from 28.49% to 62.16%.

The three isolates that exhibited the strongest inhibitory effects on *Trametes versicolor* were *Paraconiothyrium* sp. (BEN36), *Fusarium acuminatum* (BEN48), and *Paraboeremia putaminum* (BEN37), with inhibition rates of 40.27%, 35.05%, and 33.09%, respectively. For *Ganoderma gibbosum*, the isolates *Nigrograna acericola* (BEN97), *Fusarium acuminatum* (BEN48), and *Paraconiothyrium* sp. (BEN36) demonstrated the highest antifungal activity with inhibition rates of 47.51%, 42.96%, and 37.70%, respectively. When it comes to *Vanderbylia fraxinea*, *Fusarium acuminatum* (BEN48), *Paraconiothyrium brasiliense* (BEN07), and *Paraconiothyrium* sp. (BEN36) showed the highest inhibition rates of 62.16%, 59.06%, and 58.51%, respectively. These five isolates, BEN07, BEN36, BEN37, BEN48, and BEN97, were selected for the culture filtrate test.

### 3.3. Inhibitory Activity of Culture Filtrate Against Wood Decay Fungi

The culture filtrate of endophytic fungi displayed significant differences in antifungal activity against *Trametes versicolor*. Notably, isolate BEN48 exhibited the highest inhibition rate, with values of 80.5% under heated conditions and 83.5% under unheated conditions, showing no significant difference between the two (Figure 3). Isolate BEN37 had the second highest antifungal activity, with inhibition rates of 33.4% when heated and 38.8% when unheated. Isolate BEN07 was the only one that showed a significant difference between the heated and unheated conditions; however, its inhibition rates were lower than those of BEN48 and BEN37. Both BEN36 and BEN97 were almost entirely inactive against *Trametes versicolor*, with inhibition rates ranging from 2.2% to 3.7%.

The isolate BEN48 showed the highest inhibition rate against *Ganoderma gibbosum*, with rates of 40.3% under heated conditions and 38.8% under unheated conditions. No significant difference was observed between these two conditions (Figure 4). In contrast, the culture filtrate from BEN07 displayed a significant difference in antifungal activity against *Ganoderma gibbosum*, exhibiting inhibition rates of 5.5% under heated conditions and 27.1% under unheated conditions. Meanwhile, isolates BEN37, BEN36, and BEN97 showed lower inhibition rates, with no significant difference between the heated and unheated conditions.

The evaluation of the antifungal capability of endophytic fungi against *Vanderbylia fraxinea* indicated that isolate BEN48 demonstrated the highest inhibition rates, achieving 72.4% under heated conditions and 76.1% under unheated conditions (Figure 5). Antifungal activity varied significantly among the different isolates. In the heated condition, some isolates, such as BEN07 and BEN37, exhibited significant differences in activity, while others showed no significant differences.

### 3.4. Effects of Culture Filtrate Concentration of Endophytic Fungi on Antifungal Activity

The culture filtrate of the endophytic fungus *Fusarium acuminatum* (BEN48) was mixed with PDA medium at three different concentrations. The antifungal activity of this fungus against wood decay fungi is shown in Figure 6 and Figure 7. The antifungal activity at a concentration of 50% was greater than that at other concentrations for all three wood decay fungi. In general, the inhibition rates of *Ganoderma gibbosum* by BEN48 were lower than *Trametes versicolor* and *Vanderbylia fraxinea* at all concentrations. There was a significant difference among concentrations, and the inhibition rate significantly decreased as the concentration of the culture filtrate was reduced. In *Trametes versicolor*, the inhibition rate decreased from 96.5% at a 50% concentration to 83.7% at a 30% concentration and 50.8% at a 10% concentration. In *Ganoderma gibbosum*, the inhibition rate dropped from 64.1% at 50% concentration to 52.6% at 30% concentration and 24.5% at 10% concentration. For *Vanderbylia fraxinea*, there was a significant difference between the 50% concentration and the lower concentrations of 30% and 10%. However, no significant difference was observed between the 30% and 10% concentrations, and the inhibition rates for all three concentrations were high, measuring 92.7%, 89.9%, and 88.8%, respectively (Figure 6 and Figure 7).

## 4. Discussion

Endophytic fungi isolated from *P. yedoensis* showed high abundance and diversity. A total of 33 species were identified from healthy trees, while 31 species were found in decayed trees (Table 1, Figure 1). The frequency of endophytic fungi differed between healthy and decayed trees. Among them, *Parapyrenochaeta maryellenpeartiae* had the highest frequency, followed by *Paraboeremia putaminum* and *Paraconiothyrium brasiliense*. These endophytic fungi, which were isolated from healthy trees, also had antifungal activities against wood decay fungi (Table 2). These fungi may contribute to the overall health of the trees without wood decay fungal infections. Endophytic fungal diversity was more abundant in *Quercus serrata* than in *Quercus mongolica*. The abundance of endophytic fungi with antifungal activity was higher in resistant trees compared to those susceptible to oak wilt disease [33]. *Paraboeremia putaminum* was also isolated from the roots of licorice plants (*Glycyrrhiza uralensis*), and its colonization could alter the fungal and bacterial communities in the rhizosphere of licorice plants [34,35]. *Paraconiothyrium brasiliense* is also an endophytic fungus obtained from the fruit of bell pepper (*Capsicum annuum*) and can produce flavonoids that have a wide range of antibacterial activity [36,37,38].

The culture filtrate from *Fusarium acuminatum* (BEN48) effectively inhibited the growth of *Trametes versicolor*, *Ganoderma gibbosum*, and *Vanderbylia fraxinea*. Interestingly, the antifungal activity of this culture filtrate remained unchanged under different heating conditions (Figure 3, Figure 4 and Figure 5). In contrast, the antifungal activities of certain endophytic fungi, such as *Colletotrichum acutatum*, *Daldinia childiae*, and *Alternaria alternata*, were influenced by heating conditions [12]. Additionally, culture filtrates from *Streptomyces blastmyceticus* demonstrated a significant difference in effectiveness between heated and unheated conditions when controlling oak wilt fungus [39]. Generally, the antifungal activities of culture filtrates tend to decrease under heated conditions compared to unheated conditions [12,39]. The concentration of the culture filtrate also affected its antifungal activity against wood decay fungi, and higher concentrations resulted in greater inhibition rates (Figure 6 and Figure 7). At a 50% concentration, the culture filtrate from *Fusarium acuminatum* (BEN48) exhibited the highest inhibition rates of 96.5%, 64.1%, and 92.7% against *Trametes versicolor*, *Ganoderma gibbosum*, and *Vanderbylia fraxinea*, respectively (Figure 6). When inhibiting *Vanderbylia fraxinea*, concentrations of 30% and 10% still demonstrated high inhibition rates of 89.9% and 88.8%, respectively (Figure 6).

*Fusarium acuminatum* is a fungal pathogen causing leaf spots in *Saposhnikovia divaricata* [40] and potato dry rot [41]. Other studies indicated that *Fusarium acuminatum* caused *Fusarium* head blight in cereal grains [42,43,44]. This fungus also caused postharvest rot on stored Kiwifruit [45] and root rot on *Ophiopogon japonicus* [46] and *Astragalus membranaceus* [47]. However, several studies have indicated that *Fusarium acuminatum* exhibits various biological activities [48,49]. For instance, two metabolic products, acuminatopyrone and chlamydosporol, have been extracted from this fungus [48]. Additionally, *Fusarium acuminatum* has the ability to uptake metals and produce antioxidant and lipid enzymes [49]. Our study demonstrates that the culture filtrate of the endophytic fungus *Fusarium acuminatum* (BEN48) can be used for the biological control of wood decay fungi such as *Trametes versicolor*, *Ganoderma gibbosum*, and *Vanderbylia fraxinea*. Additionally, heating has been found to not affect the antifungal activity of the culture filtrate. Therefore, the research and development of biological products and biological pesticides from potential raw materials of *Fusarium acuminatum* (BEN48) is necessary and should be carried out in further studies.

## 5. Conclusions

The diversity of endophytic fungi isolated from *P. yedoensis* varied between decayed and healthy trees. A total of 31 species were isolated from decayed trees, while 33 species were obtained from healthy trees. The healthy trees exhibited a greater abundance of endophytic fungi that were effective in mycelial inhibition against wood decay fungi compared to the decayed trees. The culture filtrate of *Fusarium acuminatum* (BEN48) demonstrated the highest inhibition rates against wood decay fungi, specifically *Trametes versicolor*, *Ganoderma gibbosum*, and *Vanderbylia fraxinea*. Additionally, heating did not affect the antifungal activity of the culture filtrate from BEN48. The concentration of this culture filtrate had significant effects on *Trametes versicolor* and *Ganoderma gibbosum*. In the case of *Vanderbylia fraxinea*, a significant difference was observed between the 50% concentration and the lower concentrations of 30% and 10%. Notably, the inhibition rates across all concentrations exceeded 88%. Therefore, *Fusarium acuminatum* (BEN48) shows potential for developing a biofungicide to control wood decay fungi and other plant pathogens.

## Figures and Tables

**Figure 1 microorganisms-13-00617-f001:**
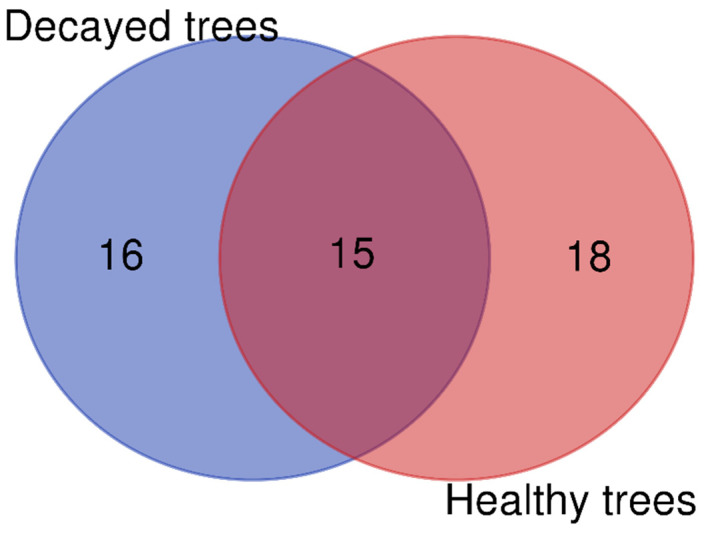
Venn diagram illustrating number of endophytic fungal isolates overlapped and non-overlapped from decayed and healthy *P. yedoensis* trees.

**Figure 2 microorganisms-13-00617-f002:**
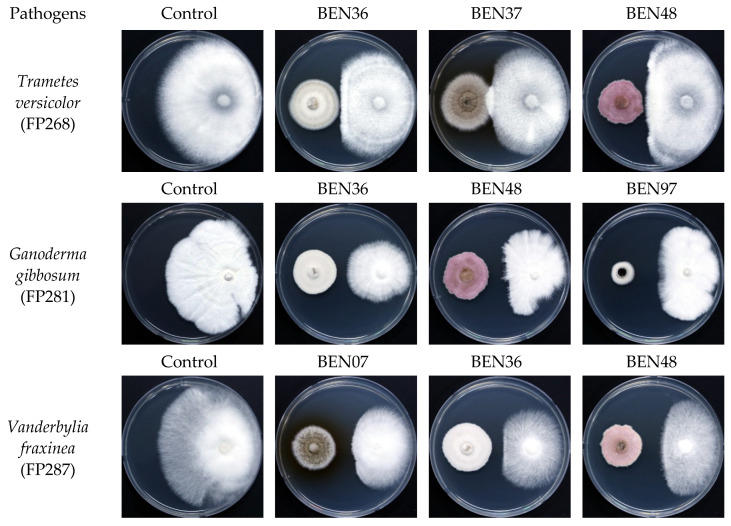
Antifungal activity of endophytic fungal isolates (**left**) against wood decay fungi (**right**) in dual culture assay. BEN07: *Paraconiothyrium brasiliense*; BEN36: *Paraconiothyrium* sp.; BEN37: *Paraboeremia putaminum*; BEN48: *Fusarium acuminatum*; BEN97: *Nigrograna acericola*.

**Figure 3 microorganisms-13-00617-f003:**
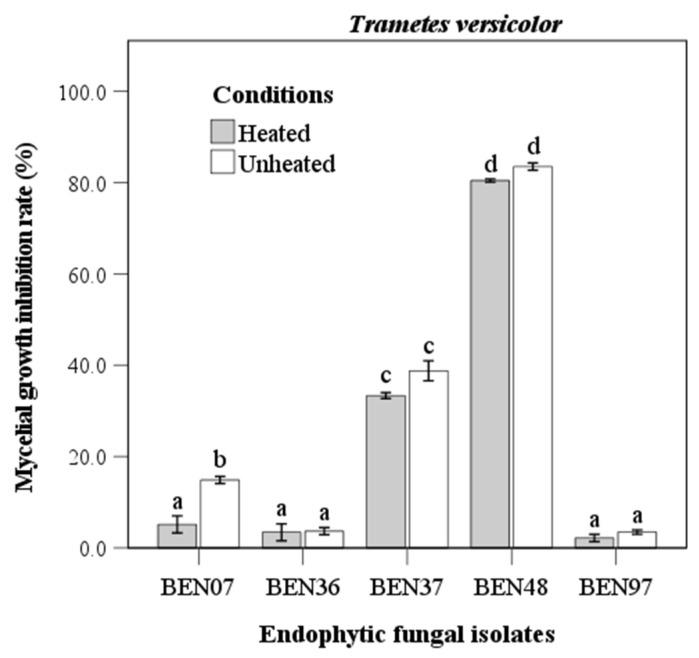
Antifungal activity of heated and unheated culture filtrates from endophytic fungal isolates against *Trametes versicolor*. Bars represent standard error of mean (n = 5). Bars labeled with same letter are not significantly different between temperature treatments at *p* < 0.05 using Tukey’s honestly significant difference test.

**Figure 4 microorganisms-13-00617-f004:**
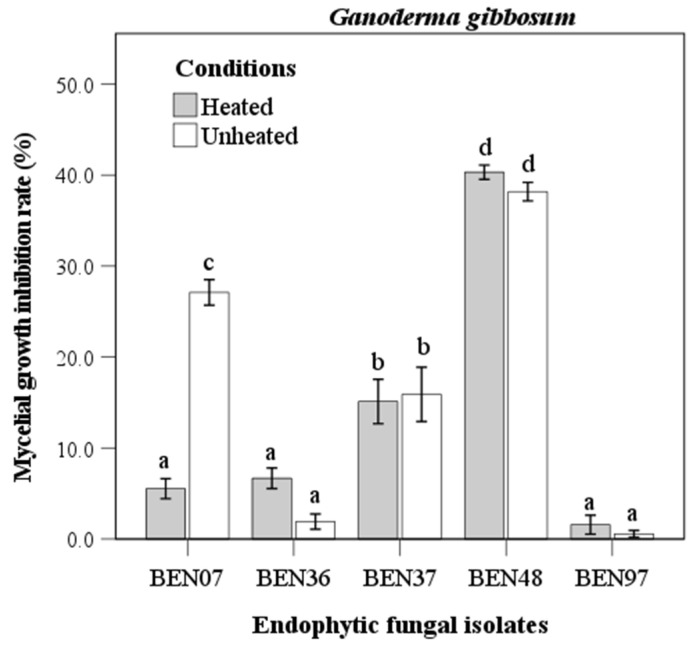
Antifungal activity of heated and unheated culture filtrates from endophytic fungal isolates against *Ganoderma gibbosum*. Bars represent standard error of mean (n = 5). Bars labeled with same letter are not significantly different between temperature treatments at *p* < 0.05 using Tukey’s honestly significant difference test.

**Figure 5 microorganisms-13-00617-f005:**
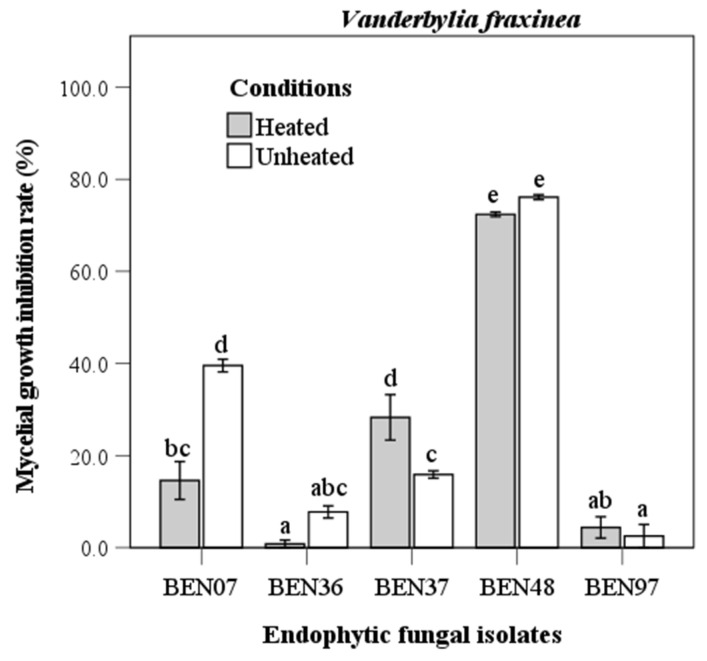
Antifungal activity of heated and unheated culture filtrates from endophytic fungal isolates against *Vanderbylia fraxinea*. Bars represent standard error of mean (n = 5). Bars labeled with same letter are not significantly different between temperature treatments at *p* < 0.05 using Tukey’s honestly significant difference test.

**Figure 6 microorganisms-13-00617-f006:**
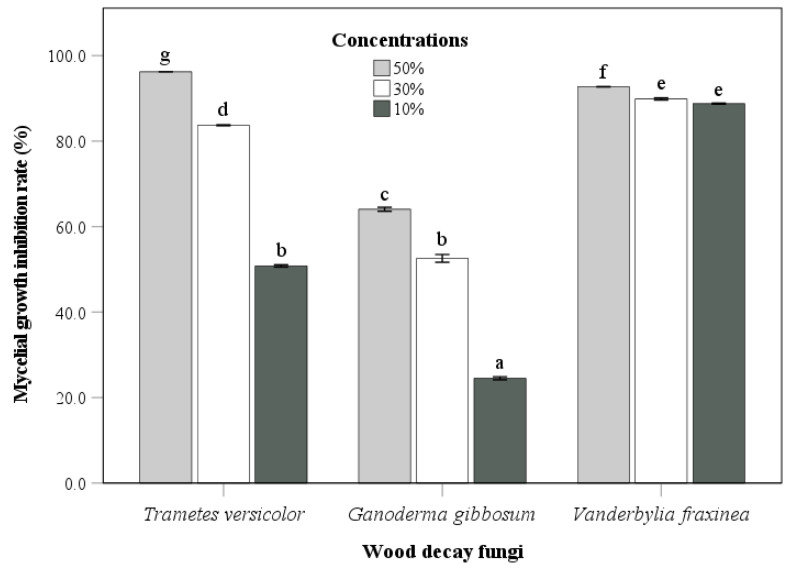
Antifungal activity of endophytic fungus *Fusarium acuminatum* (BEN48) against three wood decay fungi at different concentrations of culture filtrates. Bars labeled with same letter are not significantly different between temperature treatments at *p* < 0.05 using Tukey’s honestly significant difference test.

**Figure 7 microorganisms-13-00617-f007:**
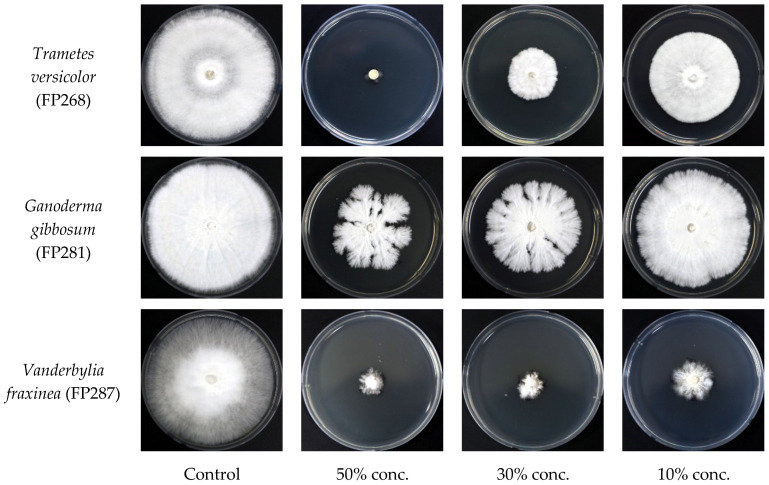
Mycelial growth of wood decay fungi on PDA medium amended with different concentrations of culture filtrate from endophytic fungus *Fusarium acuminatum* (BEN48).

**Table 1 microorganisms-13-00617-t001:** Diversity and frequency comparisons of endophytic fungi isolated from decayed and healthy *P. yedoensis* trees.

Isolate No.	Most Closely Related Strain	GenBank Acc. No.	Maximum Identity (%)	Decayed Trees		Healthy Trees	
				No. of Isolates	Frequency (%)	No. of Isolates	Frequency (%)
1	*Allophoma zantedeschiae*	MH855298	100.0	0	0.00	1	1.37
2	*Alternaria alternata*	OP596149	100.0	1	0.54	3	4.11
3	*Aspergillus aculeatus*	MT541884	99.8	2	1.09	0	0.00
4	*Aspergillus sydowii*	MN413179	100.0	1	0.54	0	0.00
5	*Candida parapsilosis*	MH545914	100.0	16	8.70	1	1.37
6	*Candolleomyces candolleanus*	MW740407	99.4	0	0.00	1	1.37
7	*Cercospora zebrina*	KM979960	100.0	0	0.00	1	1.37
8	*Chromolaenicola* sp.	OP058968	99.8	0	0.00	1	1.37
9	*Cladosporium perangustum*	MT466522	100.0	1	0.54	0	0.00
10	*Cladosporium ramotenellum*	MT223790	100.0	1	0.54	0	0.00
11	*Didymella microchlamydospora*	ON712374	100.0	1	0.54	1	1.37
12	*Effuseotrichosporon vanderwaltii*	NR_153975	99.8	1	0.54	0	0.00
13	*Fusarium acuminatum*	MT566456	100.0	0	0.00	2	2.74
14	*Fusarium oxysporum*	KU059956	100.0	1	0.54	1	1.37
15	*Fusarium parceramosum*	OR069698	100.0	0	0.00	3	4.11
16	*Hawksworthiomyces taylorii*	NR_155176	99.5	3	1.63	0	0.00
17	*Jattaea mookgoponga*	MN809156	99.5	1	0.54	0	0.00
18	*Monocillium* sp.	MG826985	88.2	0	0.00	1	1.37
19	*Nakazawaea ishiwadae*	KY104370	100.0	3	1.63	0	0.00
20	*Nigrograna acericola*	NR_190272	100.0	1	0.54	0	0.00
21	*Paraboeremia putaminum*	MK601233	100.0	22	11.96	1	1.37
22	*Paraconiothyrium brasiliense*	OP178958	100.0	17	9.24	7	9.59
23	*Paraconiothyrium* sp.	MN105575	99.8	0	0.00	3	4.11
24	*Paraphaeosphaeria sporulosa*	KY977581	98.2	0	0.00	1	1.37
25	*Parapyrenochaeta maryellenpeartiae*	OQ297061	99.3	66	35.87	16	21.92
26	*Parengyodontium album*	MT626052	99.7	1	0.54	1	1.37
27	*Penicillium camemberti*	MT529919	100.0	1	0.54	0	0.00
28	*Penicillium citrinum*	MT529135	100.0	9	4.89	2	2.74
29	*Penicillium cremeogriseum*	KU933446	99.5	0	0.00	1	1.37
30	*Penicillium paneum*	LC133775	99.8	1	0.54	0	0.00
31	*Penicillium rubens*	LT558874	99.8	2	1.09	0	0.00
32	*Penicillium sumatraense*	OQ332391	99.6	10	5.43	1	1.37
33	*Pestalotiopsis vismiae*	OR135786	100.0	1	0.54	2	2.74
34	*Phaeoacremonium iranianum*	MG745842	100.0	2	1.09	1	1.37
35	*Phaeoacremonium scolyti*	MN368438	99.6	0	0.00	1	1.37
36	*Pleurostomophora richardsiae*	AB364695	99.8	0	0.00	1	1.37
37	*Pseudeurotium bakeri*	MN518429	100.0	0	0.00	1	1.37
38	*Pseudopithomyces sacchari*	ON207654	100.0	0	0.00	1	1.37
39	*Punctularia strigosozonata*	AB374288	98.5	0	0.00	1	1.37
40	*Purpureocillium lilacinum*	MT530235	100.0	2	1.09	0	0.00
41	*Pyrenochaeta* sp.	MZ380129	100.0	1	0.54	6	8.22
42	*Pyrenochaetopsis* sp.	MZ380130	100.0	0	0.00	3	4.11
43	*Rhodotorula mucilaginosa*	MT465994	99.7	1	0.54	2	2.74
44	*Setophaeosphaeria badalingensis*	KJ869162	100.0	1	0.54	0	0.00
45	*Sphaerostilbella aureonitens*	MK838858	98.6	1	0.54	3	4.11
46	*Talaromyces diversus*	LT558943	100.0	1	0.54	0	0.00
47	*Teichospora* sp.	PP060668	99.2	0	0.00	1	1.37
48	*Trichoderma citrinoviride*	KY750460	100.0	12	6.52	0	0.00
49	*Zopfiella tabulata*	AY999132	97.7	0	0.00	1	1.37
Total				184	100.0	73	100.0

**Table 2 microorganisms-13-00617-t002:** Mycelial growth inhibition rate (%) of endophytic fungi isolates against wood decay fungi in dual culture assay.

No.	Isolates	Endophytic Fungi	Host Tree	Mycelial Growth Inhibition Rate (%) *		
				FP268 *(Trametes versicolor)*	FP281 *(Ganoderma gibbosum)*	FP287 *(Vanderbylia fraxinea)*
1	BEN04	*Pyrenochaeta* sp.	Healthy tree	24.50 ^cd^	19.61 ^abc^	39.58 ^cd^
2	BEN06	*Teichospora* sp.	Healthy tree	12.59 ^a^	12.50 ^a^	28.87 ^ab^
3	BEN07	*Paraconiothyrium brasiliense*	Healthy tree	22.55 ^bcd^	29.64 ^cde^	**59.06** ^f^
4	BEN35	*Parapyrenochaeta maryellenpeartiae*	Healthy tree	15.72 ^ab^	14.80 ^ab^	35.29 ^bc^
5	BEN36	*Paraconiothyrium* sp.	Healthy tree	**40.27** ^f^	**37.70** ^efg^	**58.51** ^ef^
6	BEN37	*Paraboeremia putaminum*	Healthy tree	**33.99** ^ef^	34.25 ^def^	52.29 ^e^
7	BEN38	*Chromolaenicola* sp.	Healthy tree	24.15 ^cd^	19.10 ^abc^	39.48 ^cd^
8	BEN48	*Fusarium acuminatum*	Healthy tree	**35.05** ^ef^	**42.96** ^fg^	**62.16** ^f^
9	BEN79	*Candolleomyces candolleanus*	Healthy tree	28.16 ^de^	26.12 ^bcd^	35.77 ^c^
10	BEN97	*Nigrograna acericola*	Decayed tree	19.04 ^abc^	**47.51** ^g^	44.61 ^d^
11	BEN104	*Monocillium* sp.	Healthy tree	19.59 ^abc^	22.88 ^abcd^	28.49 ^a^

* The average inhibition rates in the same column followed by the same letter are not significantly different from each other (*p* > 0.05) using Tukey’s honestly significant difference test. The bolded numbers indicate the MGI of the potential isolates that were used for the culture filtrate test.

## Data Availability

The data presented in this study are available on request from the corresponding authors. The data are not publicly available due to institutional policy.

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
