# Peer review of "Diversity of Endophytic Fungi Isolated from Prunus yedoensis and Their Antifungal Activity Against Wood Decay Fungi"

_microorganisms, 2025, doi:10.3390/microorganisms13030617_

Round 1

Reviewer 1 Report

Comments and Suggestions for Authors

This is a valuable manuscript. The introduction is written in an interesting way because it touches upon various aspects concerning endophytes. In the Materials & Methods section, most of the data is clearly formulated. However, some data require supplementation (see Remarks). It requires a clear statement of the fact: whether the samples from diseased trees concerned living tissue. If this wood showed wood rot or was dead or discolored - it means that the results cannot be interpreted in such a way that endophytes were isolated. Endophytes are  defined as microorganisms that spend whole or at least part of their life cycle in living plant tissue without causing disease symptoms. This is a very important problem that must be clearly articulated.

Isolations of fungi were made from decayed and healthy Prunus yedoensis trees. 257 fungal isolates were obtained. On the basis of morphology and molecular analysis, 31 fungal species in decayed trees, and 33 species in healthy trees were identified. The ability of fungi / endophytes and their culture filtrates to inhibit selected three wood rot fungi was assessed: Trametes versicolor, Ganoderma gibbosum, and Vanderbylia fraxinea. It has been assessed that Fusarium acuminatum shows potential for developing a biofungicide to control wood decay fungi. These are valuable results that, after minor revision, deserve to be published in Microorganisms.

Remarks

Line 82 it should be given from how many wood fragments the isolations were made

Line 119  it should be given from where the isolates of the three wood decay fungi: Ganoderma gibbosum, Trametes versicolor, and Vanderbylia fraxinea came. If possible, provide the Accession numbers in Genbank

Line 129 "Mycelial growth inhibition (MGI) of endophytic fungi" - this is probably an error, because the text given below (line 131) concerns the inhibition of the pathogen

Line `139 both heated and unheated - provide details

Line 170 in Table 1. change "No. of isolate" to 'No. of isolates'

Line 185 Table 2 should explain what bold means

Line 269 -271 "and all of them were isolated from healthy trees …..." - it is not clear what the authors mean. All three of these species were many times more often isolated from diseased trees. Paraboeremia putaminum - has been isolated only once from healthy trees. This text requires change or different interpretation.

Line 263 Discussion - in this section you should discuss how disease of some parts of the tree may affect endophytes in parts of the tree that are not affected by the disease. This aspect was not addressed at all in the Discussion

Author Response

Responses to Reviewer 1

We are grateful for the constructive evaluations by the reviewer and for the helpful comments. In the following, changes between the previous and the new submission are listed.

General reviewer comment:

This is a valuable manuscript. The introduction is written in an interesting way because it touches upon various aspects concerning endophytes. In the Materials & Methods section, most of the data is clearly formulated. However, some data require supplementation (see Remarks). It requires a clear statement of the fact: whether the samples from diseased trees concerned living tissue. If this wood showed wood rot or was dead or discolored - it means that the results cannot be interpreted in such a way that endophytes were isolated. Endophytes are defined as microorganisms that spend whole or at least part of their life cycle in living plant tissue without causing disease symptoms. This is a very important problem that must be clearly articulated.

Response: we collected samples in the living tissue of decayed trees determined by a sonic tomograph, not in decayed tissue. Hence, fungi from the samples were endophytic microorganisms.

Isolations of fungi were made from decayed and healthy Prunus yedoensis trees. 257 fungal isolates were obtained. On the basis of morphology and molecular analysis, 31 fungal species in decayed trees, and 33 species in healthy trees were identified. The ability of fungi / endophytes and their culture filtrates to inhibit selected three wood rot fungi was assessed: Trametes versicolor, Ganoderma gibbosum, and Vanderbylia fraxinea. It has been assessed that Fusarium acuminatum shows potential for developing a biofungicide to control wood decay fungi. These are valuable results that, after minor revision, deserve to be published in Microorganisms.

Thanks, no changes were made.

Detailed reviewer comment #1

Line 82 it should be given from how many wood fragments the isolations were made

Response: We added some text with red color in the revised manuscript at line 86.

Line 119  it should be given from where the isolates of the three wood decay fungi: Ganoderma gibbosum, Trametes versicolor, and Vanderbylia fraxinea came. If possible, provide the Accession numbers in Genbank.

Response: Three wood decay fungi were isolated from Prunus yedoensis. We added some text with red color in the revised manuscript at line 122-125.

Line 129 "Mycelial growth inhibition (MGI) of endophytic fungi" - this is probably an error, because the text given below (line 131) concerns the inhibition of the pathogen

Response: We included red text in the revised manuscript at line 135 to provide more details.

Line `139 both heated and unheated - provide details

Response: heated condition means the culture filtrate was autoclaved at 121oC for 15 mins. We add some red text in the revised manuscript at line 145-146.

Line 170 in Table 1. change "No. of isolate" to 'No. of isolates'

Response: we already changed according your comments in the table 1.

Line 185 Table 2 should explain what bold means

Response: we explained and added some text in the revised manuscript at line 192.

Line 269 -271 "and all of them were isolated from healthy trees …..." - it is not clear what the authors mean. All three of these species were many times more often isolated from diseased trees. Paraboeremia putaminum - has been isolated only once from healthy trees. This text requires change or different interpretation.

Response: Parapyrenochaeta maryellenpeartiae, Paraboeremia putaminum and Paraconiothyrium brasiliense were isolated from both healthy and decayed trees. However, the isolates from decayed trees did not show antifungal activity in the dual culture test. We added some red text to describe clearer at the line 275.

Line 263 Discussion - in this section you should discuss how disease of some parts of the tree may affect endophytes in parts of the tree that are not affected by the disease. This aspect was not addressed at all in the Discussion.

Response: Our study only compared between decayed trees and healthy trees. In the same tree, we did not compared different parts. For further study, we will compare different parts of tree and discuss about this.

Reviewer 2 Report

Comments and Suggestions for Authors

Article: Diversity of endophytic fungi isolated from Prunus yedoensis and their antifungal activity against wood decay fungi.

In this work, endophytic fungi of Prunus yedoensis trees were investigated. A total of 31 species were isolated, 23 from decayed trees, while 33 species were obtained from healthy trees. Their ability to inhibit the growth of wood decay fungi was analyzed. Overall, the work is relevant and well planned, the manuscript is written clearly and logically, the data are presented well. However, I have some comments.

  1. Line 99. What microscope was used?
  2. Line 113. Specify the manufacturer of the sequencer.
  3. The full species name is given only at the first mention in the text, then the abbreviated version is used (P. yedoensis).
  4. In which appendix were figures 3,4,5 obtained?
  5. The authors used ITS sequences to identify the fungi. It remains unclear whether ITS1 or ITS2 is used. It is worth noting that it is impossible to accurately determine a number of fungi to the species level based on the ITS sequence. For example, several marker genes are used to identify Alternaria species. In some cases, up to 6. If we perform a Blust of the given ITS sequence, we will find that it is 100% identical to A. tenuissima and A. infectoria. So in this case, the authors should use the generic name Alternaria sp.. Moreover, this remark also applies to other fungi, such as Aspergiluss and others. It is necessary to review all species names and correct them where necessary.
  6. Moreover, the authors provide in the text and Table 1 the numbers of the GenBank sequences obtained in other works, but do not indicate this. The nucleotide sequences obtained in this work are also not presented.

Author Response

Responses to Reviewer 2

We are grateful for the constructive evaluations by the reviewer and for the helpful comments. In the following, changes between the previous and the new submission are listed.

General reviewer comment:

In this work, endophytic fungi of Prunus yedoensis trees were investigated. A total of 31 species were isolated, 23 from decayed trees, while 33 species were obtained from healthy trees. Their ability to inhibit the growth of wood decay fungi was analyzed. Overall, the work is relevant and well planned, the manuscript is written clearly and logically, the data are presented well. However, I have some comments.

Thanks, no changes were made.

Detailed reviewer comment

Line 99. What microscope was used?

Response: We added information of microscope in the revised manuscript at line 102.

Line 113. Specify the manufacturer of the sequencer.

Response: We added information of manufacturer of the sequencer at line 116 in the revised manuscript.

The full species name is given only at the first mention in the text, then the abbreviated version is used (P. yedoensis).

Response: We changed the abbreviated version of P. yedoensis in the the revised manuscript at line 63, 75, 83, 161, 162, 177, 180, 270, 315.

In which appendix were figures 3,4,5 obtained?

Response: We prefer to use the full scientific names in figures for easier reading and understanding of the information presented. No changes were made.

The authors used ITS sequences to identify the fungi. It remains unclear whether ITS1 or ITS2 is used. It is worth noting that it is impossible to accurately determine a number of fungi to the species level based on the ITS sequence. For example, several marker genes are used to identify Alternaria species. In some cases, up to 6. If we perform a Blust of the given ITS sequence, we will find that it is 100% identical to A. tenuissima and A. infectoria. So in this case, the authors should use the generic name Alternaria sp.. Moreover, this remark also applies to other fungi, such as Aspergiluss and others. It is necessary to review all species names and correct them where necessary.

Response:  We edited and compared carefully the sequences of our samples. The identifications were based on sequencing results of ITS1 and ITS2 sequences using the primer set ITS1F and ITS4. Our results were determined based on ITS1, 5.8s, and ITS2 sequences, which we already described in the methods of the study. We totally understand what you mean about the fungal identification using single ITS region sequences. However, we were trying to find fungal endophytes to inhibit the growth of the pathogens, wood decay fungi, in the paper, not for the identification or systematic study of the endophytic fungi. So, the information on endophytic fungi identified with the most closely related strain is enough for this study. For further study, if we continue to deeply research species of given endophytic fungi effective against wood decay fungi, we will use multiple genes to confirm the species' names in the next manuscript.

Moreover, the authors provide in the text and Table 1 the numbers of the GenBank sequences obtained in other works, but do not indicate this. The nucleotide sequences obtained in this work are also not presented.

Response: The GenBank sequences we used had the closest identification with our samples. We also had nucleotide sequences of all isolates that can be sent to reviewers if necessary. We will deposit them for the identification and systematic study in the next manuscript.

Reviewer 3 Report

Comments and Suggestions for Authors

The manuscript microorganisms-3489187, titled “Diversity of endophytic fungi isolated from Prunus yedoensis and their antifungal activity against wood decay fungi”. The study found that healthy trees contained a higher number of endophytic fungi with antifungal activity. Fusarium acuminatum (BEN48) exhibited the highest inhibitory effects against Trametes versicolor, Ganoderma gibbosum, and Vanderbylia fraxinea, suggesting its potential as a biocontrol agent for managing wood decay fungi.  However, in my opinion this paper must be revised in a major manner for reasons of forms and content.

Introduction

Could the authors elaborate on the function of endophytic fungi in defending trees against fungi that cause wood decay? How do they compare to other biocontrol strategies?

Could the authors explain why P. yedoensis was selected as the study's host plant? Is there a particular economic or ecological reason?

The study indicates that there is little research being done in South Korea on managing wood rot fungi. Are there any limitations to the existing control methods that the authors could elaborate on?

Materials and Methods

How was it standardised to choose both good and unhealthy trees? Did genetic or environmental factors play a role?

Could the authors explain why they decided to use ethanol and NaOCl as the particular sterilisation method? Were alternate methods of sterilisation taken into account?

Before analysis, were the fungal isolates kept in storage for an extended period of time? If yes, what measures were taken to ensure their viability?

What method was used to identify the fungal isolates at the species level? Were closely related species able to be distinguished using molecular identification (ITS sequencing)?

Results

The study indicates a higher diversity of endophytic fungi in healthy trees compared to dead ones. Could the writers talk about potential ecological explanations for this discrepancy?

Could the authors elaborate on if any fungi from dead trees have antifungal properties? If not, what might be the cause of this?

Different concentrations of culture filtrates were used to investigate their antifungal activity. Did the authors look at how different fungal isolates might work together?

Discussion

According to the study, Fusarium acuminatum (BEN48) may be used as a biocontrol agent. Are there any worries that plants could be harmed by Fusarium species?

What is the difference between the commercial antifungal drugs and the inhibition rates of the detected fungi? The impact of heating on antifungal activity is covered in the study.

Could the authors make any educated guesses about the type of active substances causing inhibition? The findings show that different fungi have different inhibition rates.

Could the authors discuss the potential processes that might be responsible for these variations?

Conclusion

The conclusion indicates that Fusarium acuminatum (BEN48) has potential for biocontrol applications. What would be the next steps for validating this in real-world conditions?

Could the authors discuss potential limitations of the study and suggest future directions for research?

Author Response

Responses to Reviewer 3

The manuscript microorganisms-3489187, titled “Diversity of endophytic fungi isolated from Prunus yedoensis and their antifungal activity against wood decay fungi”. The study found that healthy trees contained a higher number of endophytic fungi with antifungal activity. Fusarium acuminatum (BEN48) exhibited the highest inhibitory effects against Trametes versicolor, Ganoderma gibbosum, and Vanderbylia fraxinea, suggesting its potential as a biocontrol agent for managing wood decay fungi.  However, in my opinion this paper must be revised in a major manner for reasons of forms and content.

Introduction

Could the authors elaborate on the function of endophytic fungi in defending trees against fungi that cause wood decay? How do they compare to other biocontrol strategies?

Response: Wood-decay fungi can initially exist in trees as endophytes. Under favorable conditions, they can act as agents of decay. However, the community of endophytic fungi includes not only potential wood-decaying species but also many other fungal species. Additionally, the interactions among these endophytes influence their overall function. As a result, decay fungi may be inhibited or may transition into wood-decomposing agents.

Utilizing endophytic fungi provides an environmentally friendly solution that eliminates the need for pesticides. Some endophytic fungi can directly inhibit pathogens and enhance the resistance and growth of their host plants.

Could the authors explain why P. yedoensis was selected as the study's host plant? Is there a particular economic or ecological reason?

Response:  P. yedoensis is commonly cultivated in urban areas in South Korea. Many decayed trees may die while standing or could fall, posing risks in urban areas.

We added some red text in the revisied manucript at line 70

The study indicates that there is little research being done in South Korea on managing wood rot fungi. Are there any limitations to the existing control methods that the authors could elaborate on?

Response: While research has been conducted on using actinomycetes strains for the biocontrol of brown rot fungi, there has been no mention of their efficacy against white rot fungi. Therefore, this study is necessary.

We added some red text in the revisied manucript at line 71-72.

Materials and Methods

How was it standardised to choose both good and unhealthy trees? Did genetic or environmental factors play a role?

Response: We mentioned about this information in our manuscript at line 84-87.

Could the authors explain why they decided to use ethanol and NaOCl as the particular sterilisation method? Were alternate methods of sterilisation taken into account?

Response: This fundamental method has been utilized in numerous studies, combining ethanol and NaOCl to ensure that no microorganisms remain on the surfaces of the samples.

Before analysis, were the fungal isolates kept in storage for an extended period of time? If yes, what measures were taken to ensure their viability?

Response: we stored wood decay fungi in the agar slants at 4°C. We added some red text in the revisied manucript at line 125.

What method was used to identify the fungal isolates at the species level? Were closely related species able to be distinguished using molecular identification (ITS sequencing)?

Response: Previous studies have conducted the identification of wood decay fungi using multiple sequencing techniques, and additional information has already been included in lines 123-124.

Results

The study indicates a higher diversity of endophytic fungi in healthy trees compared to dead ones. Could the writers talk about potential ecological explanations for this discrepancy?

Response: Endophytic fungal diversity can differ across various ecological regions; however, our study focused solely on healthy and diseased plants within the same ecological region.

Could the authors elaborate on if any fungi from dead trees have antifungal properties? If not, what might be the cause of this?

Response: We have not yet tried to isolate fungi from dead trees, but decayed trees can provide endophytic fungi with antifungal properties. This is illustrated in Table 2 for the BEN97 isolate.

Different concentrations of culture filtrates were used to investigate their antifungal activity. Did the authors look at how different fungal isolates might work together?

Response: Thank you for your recommendation. We also plan to conduct experiments to test the antifungal activities of a mixture that includes several endophytic fungi. In fact, the antifungal activities can either improve or decrease when these fungi interact with each other. After completing our experiments, we will discuss the results in our future papers.

Discussion

According to the study, Fusarium acuminatum (BEN48) may be used as a biocontrol agent. Are there any worries that plants could be harmed by Fusarium species?

Response: Fusarium acuminatum has been a plant pathogen in herbaceous plants, not in woody plants yet. However, it is necessary to test its pathogenicity before using it in the field test.  Heating does not affect the antifungal activity of Fusarium acuminatum (BEN48). Therefore, we recommend heating (disinfecting) before field use, ensuring that plants are not harmed by Fusarium species. The pathogenicity test in the cherry tree will be conducted in our future study. Thanks.

What is the difference between the commercial antifungal drugs and the inhibition rates of the detected fungi? The impact of heating on antifungal activity is covered in the study.

Response: In this study, we did not compare the antifungal activities of commercial antifungal drugs with those of endophytic fungi. However, the inhibition rates from our study were notably high, ranging from 64.1% to 96.5% at a concentration of 50%. For this reason, we recommend the use of Fusarium acuminatum (BEN48) as a biocontrol agent. We will compare the culture filtrates and fungicides for future study.

Could the authors make any educated guesses about the type of active substances causing inhibition? The findings show that different fungi have different inhibition rates.

Response: The identification of compounds should be conducted using techniques in natural product chemistry. Published literature indicates that endophytic fungi can produce a variety of compounds with the ability to inhibit pathogenic fungi. In this study, our goal is to identify potential endophytic fungal strains and subsequently investigate the antifungal compounds they produce, as well as the development of commercial products.

Could the authors discuss the potential processes that might be responsible for these variations?

Response: we discussed this in the manuscript at lines 301-303.

Conclusion

The conclusion indicates that Fusarium acuminatum (BEN48) has potential for biocontrol applications. What would be the next steps for validating this in real-world conditions?

Response: The next steps will involve testing antifungal activities on trees. We will inject culture filtrate and wood decay fungi into the trees and monitor wood discoloration to evaluate the effects of endophytic fungi in the next study.

Could the authors discuss potential limitations of the study and suggest future directions for research?

Response: Using Fusarium acuminatum (BEN48) without autoclaving poses risks to plants, as it can be a plant pathogen. Therefore, heating it before use is safer and does not affect its antifungal activity. The pathogenicity test in the cherry tree will be conducted in our future study.

Round 2

Reviewer 2 Report

Comments and Suggestions for Authors

If you cannot or do not intend to reliably identify the species, then indicate the genus name followed by "sp." This way you will not mislead the reader.

Author Response

Responses to Reviewer 2

General reviewer comment:

If you cannot or do not intend to reliably identify the species, then indicate the genus name followed by "sp." This way you will not mislead the reader.

Response: We sincerely appreciate your comment. However, we prefer to use the scientific name based on ITS sequences due to our high identity values, with nearly all species showing 100% identity. As we mentioned last time, we were trying to find fungal endophytes to inhibit the growth of wood decay fungi in the paper, not for precise identification or systematic study of the endophytic fungi. So, we think that the species name should be kept in this manuscript and the reader will not mislead the contents. Several recently published articles have also relied exclusively on ITS sequences to identify endophytic fungi. Please refer to the published articles below that use only ITS sequences to identify endophytic fungi with antifungal activities.

For further study, if we continue to deeply research the species of given endophytic fungi effective against pathogens, we will use multiple genes to identify the species in the next manuscript.

  1. Zhang, Q.; Zhang, J.; Yang, L.; Zhang, L.; Jiang, D.; Chen, W.; Li, G. Diversity and Biocontrol Potential of Endophytic Fungi in Brassica napus. Biological Control 2014, 72, 98–108, doi:10.1016/j.biocontrol.2014.02.018.
  2. Abaya, A.; Xue, A.; Hsiang, T. Selection and Screening of Fungal Endophytes against Wheat Pathogens. Biological Control 2021, 154, 104511, doi:10.1016/j.biocontrol.2020.104511.
  3. Ramudingana, P.; Mamphogoro, T.P.; Kamutando, C.N.; Maboko, M.M.; Modika, K.Y.; Moloto, K.W.; Thantsha, M.S. Antagonistic Potential of Endophytic Fungal Isolates of Tomato (Solanum Lycopersicum L.) Fruits against Post-Harvest Disease-Causing Pathogens of Tomatoes: An in Vitro Investigation. Fungal Biology 2024, 128, 1847–1858, doi:10.1016/j.funbio.2024.05.006.
  4. Hussein, J.M.; Myovela, H.; Tibuhwa, D.D. Diversity of Endophytic Fungi from Medicinal Plant Oxalis Latifolia and Their Antimicrobial Potential against Selected Human Pathogens. Saudi Journal of Biological Sciences 2024, 31, 103958, doi:10.1016/j.sjbs.2024.103958.
  5. Díaz-Urbano, M.; Velasco, P.; Abilleira, R.; Poveda, J.; Soengas, P.; Rodríguez, V.M. Diversity and Biological Activity of Culturable Endophytic Fungi Isolated from Turnip (Brassica Rapa Subsp. Rapa) Roots. Scientia Horticulturae 2025, 339, 113861, doi:10.1016/j.scienta.2024.113861.